# Increasing the Salt Stress Tolerance of Some Tomato Cultivars under the Influence of Growth Regulators

**DOI:** 10.3390/plants12020363

**Published:** 2023-01-12

**Authors:** Mihaela Covașă, Cristina Slabu, Alina Elena Marta, Carmenica Doina Jităreanu

**Affiliations:** Department of Plant Science, Iasi University of Life Sciences, 3 Sadoveanu Alley, 700490 Iasi, Romania

**Keywords:** biostimulant, salt stress, tomato, chlorophyll, proline

## Abstract

Areas with saline soils are in continuous expansion, and in this context, it is very important to find solutions that help plants adapt more easily to these stress conditions, and to identify the main physiological and biochemical mechanisms involved in determining a good adaptability of plants. Biostimulants could be a plausible solution. This study was conducted in 2021 at the IULS (Iasi University of Life Sciences) in Romania, under greenhouse conditions and the biological material consisted of four tomato varieties: Buzau, Elisabeta, Bacovia, and Lillagro. For the treatments, we used natrium chloride (NaCl) 120 mM and an Atonik biostimulant. Three treatments were applied at intervals of 14 days. The Atonik biostimulant was applied by foliar spray, and the saline solution was applied to the root system. We have gathered some observations on the growth and fruiting character of the tomato plants studied: the height of the stems, the number of flowers in the inflorescence, the number of fruits, and the weight of fruits. Chlorophyll and carotenoid pigments as well as proline amino acid from leaves were also measured. Observations were made 14 days after the application of each treatment. Quantitative determinations were made 14 days after the application of the third treatment. The findings of this study made it clear that the Atonik biostimulant presented a positive effect on the physiological processes observed in tomato plants grown under salt stress conditions.

## 1. Introduction

Nowadays, saline soils are increasingly expanding globally, the main cause often being the result of inappropriate agricultural practices that overlap with aridification and desertification influenced by global climate change [1,2,3]. It has been estimated that, at the level of the entire planet, the surfaces affected by excess salinity exceed 200 million ha [4]. Saline and alkaline soils contain large amounts of easily soluble salts. The most harmful are natrium chloride (NaCl), sodium carbonate (Na_2_CO_3_) and sodium sulphate (Na_2_SO_4_) [5].

As a general characteristic, plants are unable to grow in saline soils where solution has an osmotic pressure of more than 10–12 atmospheres. The negative effect of salts on plants depends not only on the concentration of the solution, but also on the nature of the salts. Chlorine ions are more toxic to plants than sulfur ions, magnesium ions are more toxic than calcium or sodium ions, and boron ions have a harmful effect, especially on fruit trees. The tolerance of plants to salts depends on the vegetation phase, the climate, salt concentration, the ratio between different ions, the texture of the soil and the water regime of the soil [6,7].

The activity of microorganisms is greatly hindered due to the negative influence exerted by salts in the soil solution on the physical and chemical properties of the protoplasm. All these properties cause saline and alkaline soils to deteriorate from poorly productive to non-productive, which affects all the physiological processes of plants [8,9].

From a financial point of view, the remediation of saline soils is extremely expensive. For this reason, agriculture directs its attention to obtaining hybrids and varieties of plants with a high resistance to saline stress. Therefore, the identification of the most important physiological and biochemical processes of plants involved in increasing the degree of tolerance to saline stress has become one of the priorities of the agricultural sector. Biostimulants are a significant aid in this regard [10,11]. Many studies show that the use of biostimulants can improve the tolerance of certain plant species to abiotic factors (soil composition, extreme salinity, acidity, high and low temperatures, drought, pollution, humidity, rain, wind, or ultraviolet radiation). Stress caused by unfavorable stimuli can reduce harvest yields because plants respond by using their energy reserves to fight stress instead of concentrating on yielding [12,13]. Phytohormones play a vital role in increasing stress resistance. These regulate the growth, development and various other metabolic processes of plants during abiotic stress. The use of phytohormones such as abscisic acid, cytokinin, brassinosteroids, salicylic acid, and jasmonic acid is of great importance during abiotic stress [14]. Tomatoes represent one of the most valuable vegetables from a nutritional point of view because they have a very pleasant taste and a high content of vitamins, lycopene and mineral salts. These characteristics have led to the consumption of tomatoes in a wide range of forms across the world.

Studies show that tomatoes have moderate salinity tolerance [15,16,17,18], which does not allow their cultivation in areas with salty soil. Identification of salinity tolerant tomato populations (e.g., *Solanum lycopersicum* L.) is useful for their further use in breeding programs and the production of seed material for farmers. Another important aspect is represented by the identification of the biochemical and physiological mechanisms involved in the salinity tolerance of tomatoes against the background of growth stimulant usage.

In many studies, one of the main processes affected by salt stress is photosynthesis, and one of the most studied amino acids involved in increasing tolerance to salt stress is proline [19]. Numerous studies revealed that the vitamin C content from fruits can be increased significantly under the influence of salt stress as an adaptation reaction [20,21,22], which determines a qualitative growth of the fruits.

Areas with saline soils expand every year, thus it is important to find solutions that aid plants to adapt more easily to these stress conditions, and to identify the main physiological and biochemical mechanisms involved in the adaptability of plants, with the aid of biostimulants.

In this study we investigated the adaptation reactions of four tomato varieties (Buzau, Elisabeta, Bacovia and Lillagro) to salt stress (NaCl) under the action of the Atonik biostimulant. These varieties of tomatoes are grown mainly in protected areas in Romania and beyond.

We have analyzed to what extent the Atonik biostimulant stimulated the adaptation reaction of tomato plants to saline stress. To do this, we made different observations on the growth and fruiting process of the studied tomato plants. At the same time, we analyzed a series of biochemical indicators.

Regarding the tolerance of these tomato varieties to salinity, as well as the interaction of this abiotic stress factor with the Atonik biostimulant, the existing data are limited. The results obtained in this study provide important information that can enable a more precise use of these tomato varieties in lands affected by soil salinity, highlighting the effect of the Atonik biostimulant in this way.

## 2. Results

### 2.1. Effect of Atonik Biostimulant on the Growth and Fruiting Characters of Tomato Plants under the Action of Saline Stress

Regarding the height, after the first treatment no significant differences were recorded between the tomato varieties studied, or between the variants analyzed, with the exception of tomato plants from the Bacovia variety, in which significant differences were recorded between the control variant and the Atonik (AT) × 120 mM natrium chloride (NaCl) and 120 mM natrium chloride (NaCl) variants. The treatment with 120 mM of solution negatively influenced the height of the stems, the average being 79 cm compared to 110 cm in the control variant (Figure 1). Significant differences were also recorded between the Atonik variant and the AT × 120 mM NaCl and 120 mM NaCl variants. To interpret the results, the variants were compared separately for each variety.

The recorded values show that the Atonik biostimulant favorably engages in the growth process of the stem, and at the same time, gives the Bacovia variety an increased resistance to salt stress.

The second treatment produced changes in the behavior of tomato plants with regard to the average height of the stems. Significant differences were observed between the control variant, the AT × 120 mM NaCl, and 120 mM NaCl variants, as well as between plants treated with Atonik and those from the AT × 120 mM NaCl and 120 mM NaCl variants (Figure 2). The values show the negative effect that salt stress had on this biometric parameter. The same observations can be noted for the Bacovia variety. The 120 mM NaCl solution negatively influenced the height of the plants in the two varieties Buzau and Bacovia, but in the plants treated with Atonik in addition to the saline solution, a slight improvement in resistance to salt stress was observed.

The Elisabeta and Lillagro varieties did not present significant differences between variants, even after the application of the third treatment (Figure 3).

The Bacovia variety maintained the same trend as in the case of the two previously applied treatments. Plants watered with 120 mM NaCl saline grew the least in height, averaging at just 96 cm compared to 127 cm for the control variant, and 129 cm for the Atonik variant. The Buzau variety shows insignificant differences between variants. Although after the second treatment the plants from the AT × 120 mM NaCl and 120 mM NaCl groups showed significant differences compared to the control and the Atonik variant, after the third treatment these differences were no longer observed, indicating that the plants managed to adapt from this point to salt stress.

The average number of flowers per plant was affected by salt stress in all four tomato cultivars studied. In the Buzau variety, significant differences were recorded between the control variant and the 120 mM NaCl variant and between the Atonik variant and the 120 mM batch, as well as between AT × 120 mM NaCl and 120 mM NaCl (Figure 4).

Regarding the effect of the treatments on the average number of fruits per plant, significant differences can be observed both between variants and between varieties (Figure 5). Plants from the AT × 120 mM NaCl variant presented a higher number of fruits than those from the 120 mM NaCl variant, which indicates that the biostimulant used was able to help the plants to overcome this type of abiotic stress more easily, resulting in a substantial boost in production, especially in the Buzau, Elisabeta and Bacovia varieties.

The average weight of the fruits per plant varied both according to the applied treatments and to the analyzed variety. Large differences were observed between the control group and the 120 mM variant in all four varieties of tomato, the values being much lower compared with the control plants (Figure 6). These results indicate a negative effect of the saline solution on the fruit weight.

### 2.2. Effect of Atonik Biostimulant on Some Biochemical Indicators of Tomato Plants under the Action of Saline Stress

The data presented in Figure 7 highlight the fact that only the Buzau variety had a higher chlorophyll *a* content (4.62 µg/mL) in the case of saline treatments, the other three varieties registered a lower content compared with the control variant, with values ranging between 2.65 µg/mL and 4.82 µg/mL. This shows that, according to the biphasic model described by Munns [23], the plants of the Buzau variety 120 Mm NaCl variant were in the first phase, that of osmotic stress, but the other three cultivars within the same variant were in transition to the second phase, that of ion toxicity. The Atonik biostimulant produced a slight increase in the chlorophyll *a* content in the studied varieties with no significant differences between the analyzed varieties.

After the application of the three treatments, the chlorophyll *b* content recorded significant differences between the control variant and the Atonik variant in the Elisabeta, Bacovia, and Lillagro varieties. Elisabeta and Lillagro had a lower content compared with the untreated plants, and Bacovia had the highest value of chlorophyll *b* concentration (5.14 µg/mL) (Figure 8).

The concentration of carotenoids showed high values compared with control plants in variants treated with 120 mM saline solution within the Elisabeta and Bacovia varieties. For the other two varieties, Buzau and Lillagro, the concentrations of carotenoids were much lower. Atonik treatments did not produce significant differences compared to the control plants in the Buzau and Bacovia cultivars but caused lower carotenoid concentration values in the Elisabeta and Lillagro cultivars (Figure 9).

After the application of the three treatments, the proline content of the leaves was determined. The results revealed significant differences between the control group and the AT × 120 mM NaCl group from the Buzau and Lillagro varieties. Significant differences were also found between the plants treated with Atonik and those treated with AT × 120 Mm NaCl in the Buzau, Elisabeta, and Lillagro varieties (Table 1). In this situation, the highest content was registered in the Buzau variety in the case of plants where the saline treatment was combined with the biostimulant Atonik (0.186 nmol mg^−1^ FW), and the lowest value (0.030 nmol mg^−1^ FW) was registered in the Lillagro variety within the same variant (AT × 120 mM NaCl).

## 3. Discussion

Under the action of saline stress, the best growth rate of the stem was recorded in the Elisabeta and Lillagro tomato varieties, and no significant differences were recorded between the variants during the three treatments. These cultivars show high resistance to salt stress. The use of Atonik biostimulant concurrently with 120 mM saline improved the stem growth rate in the Bacovia cultivar.

The number of flowers decreased by 57.14%, 40%, and 50%, respectively. In the variants treated with 120 mM, flower abortion was observed in a higher proportion compared to the variants where the saline solution was applied simultaneously with the biostimulant. This phenomenon is frequently found in plants that are subjected to salinity stress [24].

The biostimulant Atonik stimulated the flowering process. In this variant, the number of flowers was much higher than that of the control variant, registering an increase of 87.5%. Salt stress also caused a decrease in the average number of flowers compared to the other three varieties. Comparing the AT × 120 mM NaCl and 120 mM NaCl variants, the positive effect that the biostimulant had on plants subjected to salt stress was noted.

At the same time, in the case of the Bacovia variety, an improvement in the flowering process was observed in plants treated with Atonik. The Atonik biostimulant exerted a positive effect on the plants subjected to saline stress within the Bacovia variety, the differences being significant. In the Lillagro cultivar, no significant differences were noticed between the plants of the AT × 120 mM NaCl and 120 mM NaCl variants but were found between the other variants, indicating that salinity negatively influenced the number of flowers in this variety as well.

The average number of flowers per plant within the Lillagro variety was negatively affected by saline solutions. Significant differences were observed between the control variant and the AT × 120 mM and 120 mM variants, as well as between the Atonik variant and the AT × 120 mM and 120 mM variants.

The Atonik biostimulant positively influenced the number of fruits in all four studied varieties. The plants subjected to salinity stress exhibited very significant differences compared to the control group as well as compared to the plants in the Atonik group, and those in the AT × 120 mM NaCl variant, the latter being obviously affected. The number of fruits decreased considerably as an unfortunate consequence of salt stress [25]. Significant differences were also noted between plants in the AT × 120 mM group and those treated to only 120 mM saline stress. These values show the positive effect that the Atonik stimulant had on the plants in increasing tolerance to salt stress.

The fruit weight was less influenced by the Atonik biostimulant; the data obtained in this case did not reflect significant differences between the control plants and those of the Atonik group. This observation was obvious in the Buzau and Elisabeta varieties. Lillagro had a lower average fruit weight per plant in the case of the biostimulant application compared with the control group, and in Bacovia, treatment with Atonik determined the translocation of assimilates to the fruits to a greater degree than in the case of the control plants.

Chlorophyll *a* is the only pigment of green plants capable of directly transferring its energy from the photosynthetic reaction. Chlorophyll *b* and carotenoid pigments have an important function in broadening the spectral range that is usable in photosynthesis. Other accessory pigments and carotenoids protect cells against photooxidation under the action of strong oxidants formed in some secondary reactions [26]. Making a comparison between the AT × 120 mM NaCl and 120 mM NaCl variants, it can be observed that the combination of biostimulant and saline solution caused an increase in the concentration of chlorophyll *a* in the leaves, except for in the Buzau variety. This increase can be explained using the biphasic model proposed by Munns (1993) [23]; it is possible that, under the action of the biostimulant, the plants remained in the osmotic stress phase for a longer time, giving them a greater capacity to adapt to saline stress

As in the case of the concentration of chlorophyll *a*, the plants from the AT × 120 mM variant recorded a higher content of chlorophyll *b* in all four varieties compared with the plants treated with 120 mM saline solution. The values ranged between 1.72 µg/mL and 3.23 µg/mL, a fact that configures a higher photosynthetic capacity and a higher degree of tolerance to salt stress.

By comparison with plants watered with 120 mM NaCl solution, AT × 120 mM NaCl treatments determined a higher concentration of carotenoids in plants of the Buzau variety (1.01 µg/mL) and lower values for the Elisabeta, Bacovia and Lillagro varieties. These results reflect the position of carotenoid pigments in relation to the photoprotection role they present [27]. The high values of carotenoids in plants watered with 120 mM saline solution indicated the triggering of a strong adaptation reaction to saline stress. Carotenoids act as accessory light-harvesting pigments and extend the absorption range. They are responsible for the quenching of light and can protect cells from damage caused by light and superoxide radicals [28,29].

Salt stress causes an imbalance of cellular ions, resulting in ionic toxicity (primary effect) and osmotic stress (secondary effect). High salinity induces the production of reactive oxygen species (ROS) such as superoxide radicals (O^2−^), singlet oxygen (^1^O^2^), hydrogen peroxide (H_2_O_2_) and consequently, through the Fenton reaction in plants, leads to the formation of the most toxic hydroxyl radicals (OH), which may interact with many essential macromolecules and metabolites causing cellular damage such as membrane deterioration [30].

To protect cells and tissues from oxidative damage as well as hyperosmotic stress, plants must produce low molecular weight non-enzymatic antioxidants such as proline [31]. Tomato plants from the Elisabeta and Lillagro varieties that were watered with 120 mM saline solution recorded a higher concentration of proline (0.431 nmol g^−1^ FW and 0.133 nmol g^−1^ FW, respectively) compared with the control group (0.098 nmol g^−1^ FW and 0.094 nmol g^−1^ FW, respectively). These results demonstrate that the protection mechanisms against salt stress was triggered leading to the accumulation of proline in plants and providing them with protection against salinity [32,33].

## 4. Materials and Methods

### 4.1. Plant Material, Treatments Used and the Growth Conditions

This study was carried in 2021 at the IULS (Iasi University of Life Sciences) in Romania, under greenhouse conditions. The biological material was represented by four tomato varieties: Buzau, Elisabeta, Bacovia and Lillagro purchased from a supplier of seeds on the Romanian market. These varieties are widely used in Romania, especially in protected areas. The bifactorial experience was conducted in plant pots with a capacity of 15 dm^3^ at each 10 kg of universal garden soil used. This type of soil is based on peat substrate and quality humus from mature bark. Other nutrients can be found in the soil, such as: Cd 1, Pb 100, Hg 1, As 20, Cr 100, Cu 100, Mo 5, Ni 50 and Zn 300, with the granulation of the soil being between 0.001 mm–0.002 mm. The experiment was randomized in blocks with four repetitions.

Three treatments were applied at intervals of 14 days. The first treatment was applied when the plants were in the phenophase of 7–8 true leaves, the second in the phenophase of 10–12 true leaves and the last before the appearance of the first inflorescence. The Atonik biostimulant was applied by foliar spray, and the saline solution with a concentration of 120 mM NaCl was applied to the root system. The treatments were applied one hour apart, on the same day. First, the Atonik biostimulant was applied and after an hour, the saline solution was applied. In order to apply the biostimulant, a manual vermorel of 2 L capacity was used. The technical parameters were: working pressure 0.3–0.4 MPa, flow rate 0.7–0.8 L/min., spray distance 60 mm, piston diameter 45 mm, with a conical spray head. The application of the biostimulant was carried out in isolation from the other plant pots. The plant pots were covered with a protective film during the Atonik treatments. No adjuvant was used on the Atonik solution. All treatments were carried out under greenhouse conditions in a controlled environment. At the treatment times, the temperature was 22 degrees and the relative air humidity was 60%. The work variants were organized as follows: control (plants irrigated with water), Atonik (plants sprayed on the leaves with the Atonik biostimulant), AT × 120 mM NaCl (plants sprayed on the leaves with the Atonik biostimulant and irrigated with a saline solution of 120 mM NaCl), 120 mM NaCl (plants irrigated with saline solution of 120 mM NaCl concentration). A 0.3% solution of Atonik was applied by double spraying the plants. The conical jet was applied perpendicularly to the plant in a zig-zag motion, from bottom to top and vice versa. At each watering, a 250 mL of NaCl solution was used with a concentration of 120 mM, the equivalent of 10.8 dS m^−1^ (corresponding to a moderately saline soil). Atonik is known as the oldest growth and fruiting biostimulant in the world and is intensively used by farmers in over 70 countries on 5 continents. The composition of the Atonik biostimulant is based on polyphenols, their chemical composition includes: sodium orthonitrophenolate 0.2%, sodium paranitrophenolate 0.3%, sodium 5-nitroguaiacolate 0.1%. Its unique composition has, among many others, a role in the proliferation and growth of leaves, in photosynthesis, flower fertility and fruit formation, even under biotic and abiotic stress conditions [34].

### 4.2. Biometric and Gravimetric Measurements

We made observations on the growth and fruiting character of the tomato plants: the height of the stems, the number of flowers in the inflorescence, the number of fruits, and their weight. The observations were made every 14 days after the application of the treatments.

### 4.3. Biochemical Analyses

#### 4.3.1. Quantitative Determination of Photosynthetic Pigments

The quantitative determination of chlorophyll pigments (chlorophyll *a*, chlorophyll *b*) and carotenoids was performed from 0.5 g of fresh leaves samples, using 80% acetone as a solvent. The absorbance measured with a UV/Vis spectrophotometer Specord 210 plus at 470, 646.8, and 663.2 nm (A470, A646.8, and A663.2) was used to calculate the concentrations of chlorophyll pigments in µg/mL based on Lichtenthaler’s equations [35,36].
c_a_ (chlorophyll *a*) = 12.25 × A_663.2_ − 2.79 × A_646.8_

c_b_ (chlorophyll *b*) = 21.50 × A_646.8_ − 5.10 × A_663.2_

c_a_ + c_b_ = 7.15 × A_663.2_ + 18.71 × A_646.8_

c_x+c_ (carotenes and xanthophylls) = (1000 × A_470_ − 1.82 × c_a_ − 85.02 × c_b_)/198 

#### 4.3.2. Proline Determination

Proline was determined from leaves after the application of the three treatments at the end of the experiment, using the modified method of Bates, 1973: 0.5 g of plant material was homogenized in 5 mL of 3% aqueous sulfosalicylic acid and the homogenate was centrifuged for 10 min at 5000 rpm. Then, 2 mL of the supernatant were reacted with 2 mL of ninhydrin acid and 2 mL of glacial acetic acid in a test tube for 1 h at 100 °C. The reaction was finished in an ice bath. The reaction mixture was extracted with 4 mL toluene and mixed vigorously for 15–20 s. The colored toluene aliquot was aspirated from the aqueous phase and the absorbance at 520 nm was read using a Specord 210 plus spectrophotometer, using toluene as a control [37].

The content of proline was calculated as nmol.mg^−1^ FW by using the formula:(Abs_extract−blank_)/Slope × Vol_extract_/Vol_aliquot_ × 1/FW 
where: Abs_extract_ = the absorbance of the extract, Blank (expressed as absorbance) and Slope = expressed as absorbance in nmol^−1^ are determined by linear regression, Vol_extract_ = the total volume of the extract, Vol_aliquot_ = the volume used in the assay and F.W. = fresh weight − the amount of plant material (mg) [38].

For the quantitative analyses, leaves from the middle of the stem have been used.

### 4.4. Statistical Analysis

To assess statistically significant differences between the treatments, the means were compared by one-way and two-way ANOVA. When the results were statistically significant, the Tukey multiple comparison test was used. The main difference was set to be significant at *p* < 0.05.

## 5. Conclusions

Salt stress affects physiological and biochemical processes depending on the variety of plant.

The results obtained through the analysis of biometric and gravimetric measurements as well as biochemical analyses (the concentration of photosynthetic pigments in the leaves and the proline content in the leaves) in the tomato varieties analyzed in this study showed a different tolerance to the saline solution of 120 mM. The Elisabeta and Lillagro varieties stood out as having better results.

The use of the Atonik biostimulant together with the 120 mM saline solution stimulated the concentration of chlorophyll *a* and *b* in all four studied varieties, indicating a higher photosynthetic capacity and a greater degree of tolerance to salt stress.

Regarding the proline content, the highest value was recorded in the Buzau variety AT × 120 mM NaCl variant, which proves that simultaneous usage of the biostimulant and the saline solution manages to trigger a greater plant adaptation reaction to this type of abiotic stress.

As a general conclusion, the biostimulant Atonik improved the salt stress resistance of the tomato plants studied.

## Figures and Tables

**Figure 1 plants-12-00363-f001:**
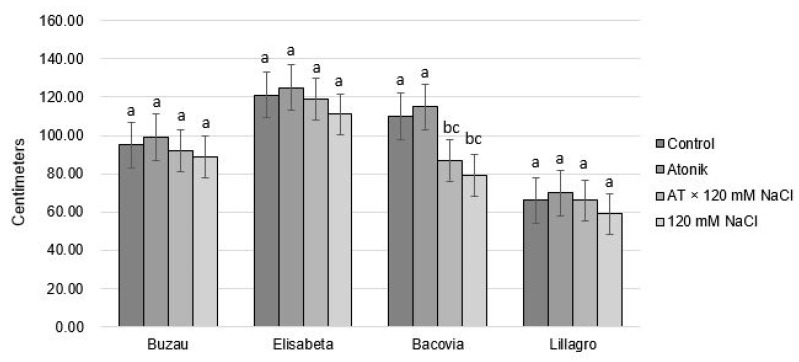
The effect of Atonik biostimulant on the height of tomato plants under saline stress after the first treatment. Different letters mean significant differences, according to the Tukey test. Error bars indicate ±SD.

**Figure 2 plants-12-00363-f002:**
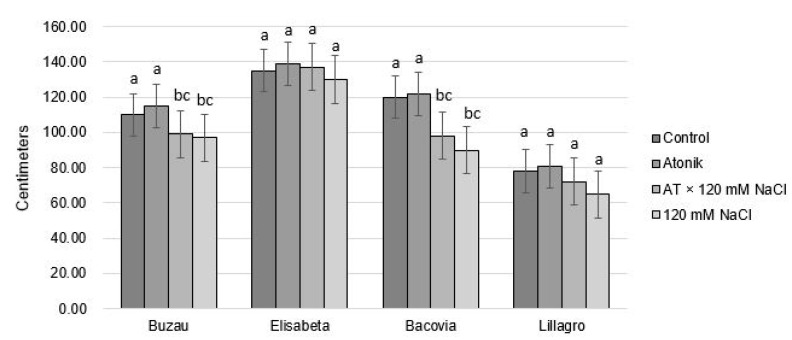
The effect of Atonik biostimulant on the height of tomato plants under saline stress after the second treatment. Different letters mean significant differences, according to the Tukey test. Error bars indicate ±SD.

**Figure 3 plants-12-00363-f003:**
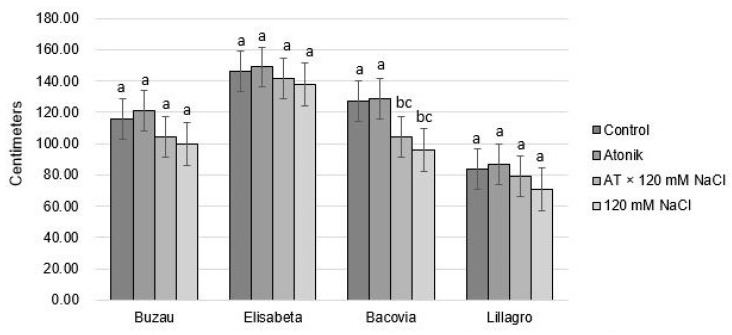
The effect of Atonik biostimulant on the height of tomato plants under saline stress after the third treatment. Different letters mean significant differences, according to the Tukey test. Error bars indicate ±SD.

**Figure 4 plants-12-00363-f004:**
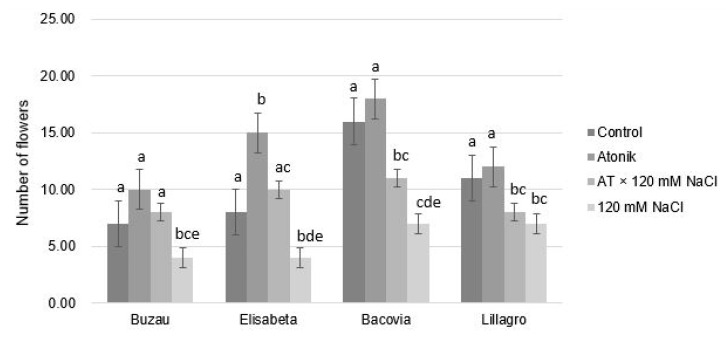
The effect of Atonik biostimulant on the number of flowers per plant under saline stress after the third treatment. Different letters mean significant differences, according to the Tukey test. Error bars indicate ±SD.

**Figure 5 plants-12-00363-f005:**
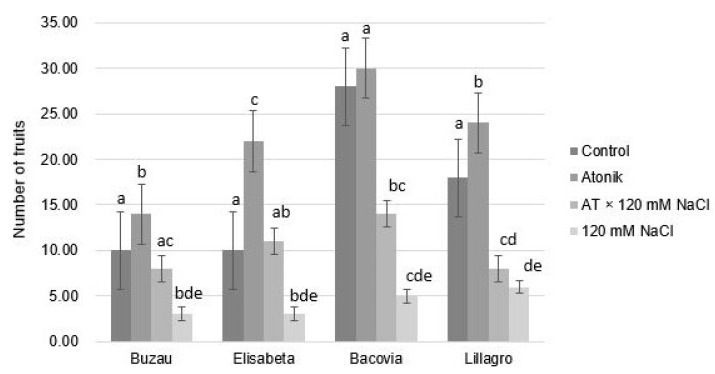
The effect of Atonik biostimulant on the number of fruits per plant under saline stress after the third treatment. Different letters mean significant differences, according to the Tukey test. Error bars indicate ±SD.

**Figure 6 plants-12-00363-f006:**
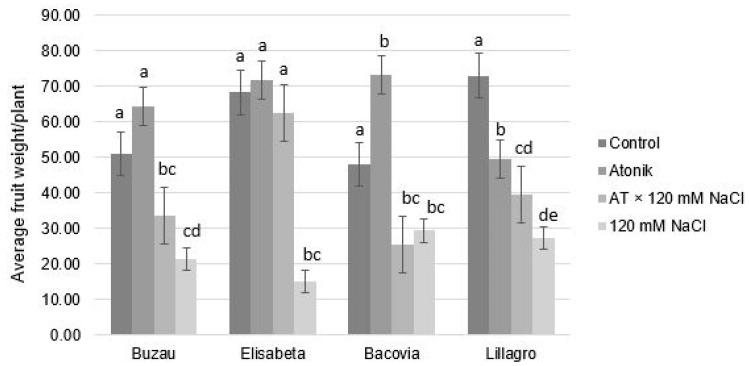
The effect of Atonik biostimulant on the average fruit weigh per plant under saline stress after the third treatment. Different letters mean significant differences, according to the Tukey test. Error bars indicate ±SD.

**Figure 7 plants-12-00363-f007:**
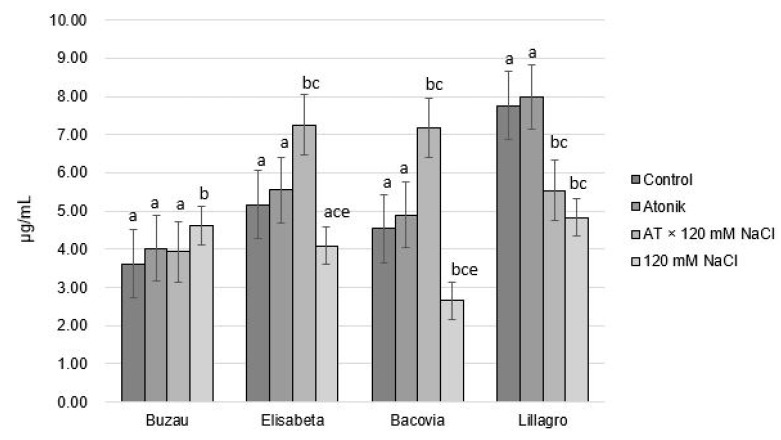
The effect of Atonik biostimulant on the concentration of chlorophyll *a* under saline stress after the third treatment. Different letters mean significant differences, according to the Tukey test. Error bars indicate ±SD.

**Figure 8 plants-12-00363-f008:**
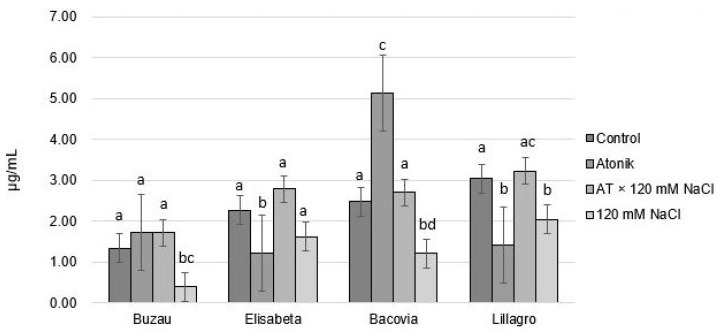
The effect of Atonik biostimulant on the concentration of chlorophyll *b* under saline stress after the third treatment. Different letters mean significant differences, according to the Tukey test. Error bars indicate ±SD.

**Figure 9 plants-12-00363-f009:**
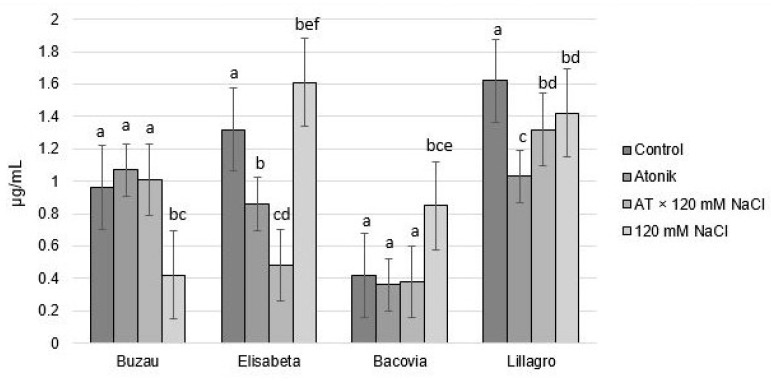
The effect of Atonik biostimulants on the concentration of carotenoids under saline stress after the third treatment. Different letters mean significant differences, according to the Tukey test. Error bars indicate ±SD.

**Table 1 plants-12-00363-t001:** Tomato leaves proline content under the action of salt stress and Atonik biostimulant. Different letters in the same column mean significant differences between treatments, according to the Tukey test (*p* ˂ 0.05).

Proline Content (nmol mg^−1^ FW) after the Third Treatments				
**Treatments**	**Buzau**	**Elisabeta**	**Bacovia**	**Lillagro**
**Control**	0.084 ^a^	0.098 ^a^	0.055 ^a^	0.094 ^a^
**Atonik**	0.084 ^a^	0.081 ^a^	0.084 ^a^	0.114 ^a^
**AT** **×** **120 mM NaCl**	0.186 ^a^	0.121 ^ab^	0.038 ^ab^	0.030 ^bc^
**120 mM NaCl**	0.084 ^a^	0.431 ^bc^	0.070 ^a^	0.133 ^b^

## Data Availability

Not applicable.

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
