# Peer review of "Increasing the Salt Stress Tolerance of Some Tomato Cultivars under the Influence of Growth Regulators"

_plants, 2023, doi:10.3390/plants12020363_

Round 1

Reviewer 1 Report

Dear Authors,

I kindly ask you to consider the observations and recommendations that I have done on the manuscript.

At the end of the Introduction section, please clear write the objective of this study.

Please pay attention to the Figures design, to the Figures titles, as well as other details.

Please prepare again the Table 1 and take into consideration the recommendations. The expression unit need to be corrected.

At the end of Materials and Methods section, please shortly describe the Statistical analysis approaches.

In the Word attached document you will find all the observations in detail.

Author Response

December, 29, 2022

Dear, Reviewer

We wish to submit the response to the comments but first we want to thank you for the your availability and kindness with which guided as to improve the article,, Increasing the salt stress tolerance of some tomato cultivars under the influence of growth regulators “ .

As you indicated, we made the following changes:

  • At the end of the Introduction section, we wrote the objective of this study.
  • We have modified the figures design, the figures titles.
  • I redid table 1 according to the recommendations.
  • At the end of Materials and Methods section, we inserted a shortly describe the Statistical analysis approaches.

All the revisions to manuscript were marked up using the “Track Changes”.

At the editor's recommendation, we made the changes in the modified manuscript from the point of view of the English language.

Please address all correspondence concerning this manuscript to me at miha_bologa@yahoo.com.

Thank you for your consideration of this manuscript.

Sincerely,

Mihaela Covașă, PhD

Assistant Professor, Department of Plant Science

IULS Iasi

0749622234

Co-author

Alina Elena Marta, PhD

Lecturer, Department of Plant Science

IULS Iasi

alinamarta_fiziologie@yahoo.com

Cristina Slabu, PhD

Lecturer, Department of Plant Science

IULS Iasi

cristinaslabu@yahoo.com

Carmenica Doina Jităreanu

Professor, Department of Plant Science

IULS Iasi

doinaj@uaiasi.ro

Reviewer 2 Report

Dear Sir/Madam

Thank you so much for your effort

The article Increasing the salt stress tolerance of some tomato cultivars under the influence of growth regulators studied. The article is well organized. But in my opinion, the article does not have enough reliable information for publishing. Because the test was done only at the greenhouse level and cannot be generalized to the field. The studies that are now published in valuable journals definitely include experiments that are related to the greenhouse and the field. Most gardens grow tomatoes in the field and not in a greenhouse. In my opinion, this article should be resubmitted to the journal by adding tests that will fix this defect. Otherwise, it should be sent to journals with a lower ranking

Author Response

December, 29, 2022

Dear, Reviewer

We wish to submit the response to the comments but first we want to thank you for the your availability and kindness with which guided as to improve the article,, Increasing the salt stress tolerance of some tomato cultivars under the influence of growth regulators “ .

These are our answers to the comments:

  • Greenhouse research from a practical point of view.

Lately, there is a tendency to grow tomatoes not only in the greenhouse, but also in other protected areas such as solariums. The idea was to support producers facing soil salinization in protected areas.

  • Greenhouse research from a scientific point of view.

Data accuracy was desired by using uniform research conditions, namely: the same electrical conductivity of the soil, the same doses of biostimulants with uniform application on the plant, irrigation with equal amounts of water.

We did not opt for research in field conditions because the salinized soils in our area are less suitable for tomato cultivation and their degree of salinization is uneven. In addition, we did not want to artificially salinize the soil by additional addition of NaCl.

The varieties used in this study are especially intended for cultivation in protected areas.

All the revisions to manuscript were marked up using the “Track Changes”.

At the editor's recommendation, we made the changes in the modified manuscript from the point of view of the English language.

Please address all correspondence concerning this manuscript to me at miha_bologa@yahoo.com.

Thank you for your consideration of this manuscript.

Sincerely,

Mihaela Covașă, PhD

Assistant Professor, Department of Plant Science

IULS Iasi

0749622234

Co-author

Alina Elena Marta, PhD

Lecturer, Department of Plant Science

IULS Iasi

alinamarta_fiziologie@yahoo.com

Cristina Slabu, PhD

Lecturer, Department of Plant Science

IULS Iasi

cristinaslabu@yahoo.com

Carmenica Doina Jităreanu

Professor, Department of Plant Science

IULS Iasi

doinaj@uaiasi.ro

Reviewer 3 Report

The researchers studied the effect of a biostimulant applied via the leaves to attenuate salt stress in different tomato cultivars. The study is very descriptive and does not add new information to what is known about the damage caused by saline stress. A novelty would be the effect of different genetic materials used. Thus, the authors should better explore this research novelty in the introduction and discussion. A critical point of the research is the lack of detailing the detailed characteristics of the biostimulant. Which polyphenol? Is it only polyphenols in the composition? Research indicates that double spraying of 0.3% Atonik solution was performed. However, details of this technique are lacking. What period between the two sprays? Was this foliar spraying performed on the same date as the saline solution was applied? How many solution was applied per plant? Was surfactant or adjuvant applied to the Atonik solution? Was the surface of the pot protected before foliar spraying? It is important to indicate the air temperature and relative humidity at the time of each foliar spray. Include data from the sprayer used, such as flow and pressure. It lacks to indicate details of the soil used as a name according to the American classification system. Include the results of the chemical analysis and also the granulometric analysis. All this detailing is very important for the authors to include in the manuscript as it deals with the main treatment of the research. Detail which leaf of the plant was collected to carry out the indicated analyses. Putting the figures together instead of 9 figures join them to form only 3 figures. For example Figure 1 would have 3 variables and so on. In the item results, the cultivars were not compared. Why? The conclusion indicates that atonik was indicated as a saline stress attenuator for all varieties. This is not correct, as figure 6 indicates that in the bacovia cultivar there is no difference in production between saline treatments with and without atonik. This is the main variable of the study and must be considered and it was necessary to discuss these aspects to indicate why this occurred. It is even important for the authors to detail the genetic differences between these cultivars in terms of material and methods. Are the parents of each cultivar similar or very different? Do the authors lack to discuss whether the biostimulant has a better benefit in tomato plants with or without saline stress? Why?

Author Response

December, 29, 2022

Dear, Reviewer

We wish to submit the response to the comments but first we want to thank you for the your availability and kindness with which guided as to improve the article,, Increasing the salt stress tolerance of some tomato cultivars under the influence of growth regulators “ .

These are our answers to the comments:

  • The material used was represented by four tomato varieties widely used in Romania, especially in protected areas. There are no studies on these tomato varieties to highlight how they react to salt stress. There are also no data on how the Atonik biostimulant stimulates the adaptation reaction of these tomato varieties to salt stress or what would be the optimal period between sprayings. At least in Romania and beyond, there are not enough studies on how biostimulants can alleviate salt stress in different horticultural species.
  • The composition of the Atonik biostimulant is based on polyphenols, chemical composition including: sodium orthonitrophenolate 0.2%, sodium paranitrophenolate 0.3%, sodium 5-nitroguaiacolate 0.1%.
  • In the study we specified that three treatments were carried out at intervals of 14 days. The first treatment was applied when the plants were in the phenophase of 7-8 true leaves, the second in the phenophase of 10-12 true leaves, and the last before the appearance of the first inflorescence.
  • The treatments were applied on the same day, one hour apart. First I applied the Atonik biostimulant and after an hour the saline solution.
  • To apply the biostimulant I used a manual vermorel 2 L capacity. The technical parameters were: working pressure 0.3-0.4 Mpa, flow rate l/mim 0.7-0.8, spray distance 60 mm, piston diameter 45 mm, conical spray head. The conical jet was applied perpendicularly to the plant in a zig-zag motion, from bottom to top and vice versa (double spray). The application of the biostimulant was carried out in isolation by the other plant pots. The plant pots were covered with a protective film during the Atonik treatments. No adjuvant was used on the Atonik solution. All treatments were carried out in greenhouse conditions, in a controlled space. At the time of applying the treatments, the temperature was 22 degrees, relative air humidity 60%.
  • We used universal garden soil based on peat substrate, quality humus from mature bark, and other soil nutrients (Cd1, Pb 100, Hg1, As 20, Cr 100, Cu 100, Mo 5, Ni 50, Zn 300) at 0.001mm - 0.002mm granulation.
  • For the quantitative analyses we used leaves from the middle of the stem.
  • In order to facilitate the interpretation of the obtained results, I made a figure for each analysis performed, except for the height of the plants, for which I realized a graph with the results obtained after the application of each treatment. We proceeded in this manner in order to clearly observe the differences between the applied treatments as well as to highlight better the duration of the transition from one treatment to another.
  • We analyzed the interaction of the biostimulant on each variety of tomato subjected to saline stress observing the reaction of each variety separately and for this reason we did not make a comparison between the varieties. This is the subject of study for another research.
  • Indeed, in the figure 6, for the Bacovia variety there are no significant differences between the variants with saline solution and saline solution and Atonik, but the other results highlight a positive effect of the biostimulant on the resistance to salt stress also for the Bacovia variety. This is reflected in figure 5 in which the number of fruits per plant is higher in the Atx120 Mm variant compared to the 120 Mm variant. The conclusion formulated at the end is a general one, as a whole, related to all the results obtained.
  • Our research is not the subject of a genetic analysis of the varieties used, perhaps in a future research we will consider this aspect.
  • The research looked at how the biostimulant Atonik influenced the tolerance of these four tomato varieties to salt stress. To see these differences, it was necessary to include in the study a variant treated with Atonik, but we were directly interested by the reaction of the tomato plants under the action of the Atonik biostimulant and the 120 Mm saline solution.

All the revisions to manuscript were marked up using the “Track Changes”.

At the editor's recommendation, we made the changes in the modified manuscript from the point of view of the English language.

The details about the application of the Atonik biostimulant were not included in the last version to the manuscript, I will do so with your permission.

Please address all correspondence concerning this manuscript to me at miha_bologa@yahoo.com.

Thank you for your consideration of this manuscript.

Sincerely,

Mihaela Covașă, PhD

Assistant Professor, Department of Plant Science

IULS Iasi

0749622234

Co-author

Alina Elena Marta, PhD

Lecturer, Department of Plant Science

IULS Iasi

alinamarta_fiziologie@yahoo.com

Cristina Slabu, PhD

Lecturer, Department of Plant Science

IULS Iasi

cristinaslabu@yahoo.com

Carmenica Doina Jităreanu

Professor, Department of Plant Science

IULS Iasi

doinaj@uaiasi.ro

December, 29, 2022

Dear, Reviewer

We wish to submit the response to the comments but first we want to thank you for the your availability and kindness with which guided as to improve the article,, Increasing the salt stress tolerance of some tomato cultivars under the influence of growth regulators “ .

These are our answers to the comments:

  • The material used was represented by four tomato varieties widely used in Romania, especially in protected areas. There are no studies on these tomato varieties to highlight how they react to salt stress. There are also no data on how the Atonik biostimulant stimulates the adaptation reaction of these tomato varieties to salt stress or what would be the optimal period between sprayings. At least in Romania and beyond, there are not enough studies on how biostimulants can alleviate salt stress in different horticultural species.
  • The composition of the Atonik biostimulant is based on polyphenols, chemical composition including: sodium orthonitrophenolate 0.2%, sodium paranitrophenolate 0.3%, sodium 5-nitroguaiacolate 0.1%.
  • In the study we specified that three treatments were carried out at intervals of 14 days. The first treatment was applied when the plants were in the phenophase of 7-8 true leaves, the second in the phenophase of 10-12 true leaves, and the last before the appearance of the first inflorescence.
  • The treatments were applied on the same day, one hour apart. First I applied the Atonik biostimulant and after an hour the saline solution.
  • To apply the biostimulant I used a manual vermorel 2 L capacity. The technical parameters were: working pressure 0.3-0.4 Mpa, flow rate l/mim 0.7-0.8, spray distance 60 mm, piston diameter 45 mm, conical spray head. The conical jet was applied perpendicularly to the plant in a zig-zag motion, from bottom to top and vice versa (double spray). The application of the biostimulant was carried out in isolation by the other plant pots. The plant pots were covered with a protective film during the Atonik treatments. No adjuvant was used on the Atonik solution. All treatments were carried out in greenhouse conditions, in a controlled space. At the time of applying the treatments, the temperature was 22 degrees, relative air humidity 60%.
  • We used universal garden soil based on peat substrate, quality humus from mature bark, and other soil nutrients (Cd1, Pb 100, Hg1, As 20, Cr 100, Cu 100, Mo 5, Ni 50, Zn 300) at 0.001mm - 0.002mm granulation.
  • For the quantitative analyses we used leaves from the middle of the stem.
  • In order to facilitate the interpretation of the obtained results, I made a figure for each analysis performed, except for the height of the plants, for which I realized a graph with the results obtained after the application of each treatment. We proceeded in this manner in order to clearly observe the differences between the applied treatments as well as to highlight better the duration of the transition from one treatment to another.
  • We analyzed the interaction of the biostimulant on each variety of tomato subjected to saline stress observing the reaction of each variety separately and for this reason we did not make a comparison between the varieties. This is the subject of study for another research.
  • Indeed, in the figure 6, for the Bacovia variety there are no significant differences between the variants with saline solution and saline solution and Atonik, but the other results highlight a positive effect of the biostimulant on the resistance to salt stress also for the Bacovia variety. This is reflected in figure 5 in which the number of fruits per plant is higher in the Atx120 Mm variant compared to the 120 Mm variant. The conclusion formulated at the end is a general one, as a whole, related to all the results obtained.
  • Our research is not the subject of a genetic analysis of the varieties used, perhaps in a future research we will consider this aspect.
  • The research looked at how the biostimulant Atonik influenced the tolerance of these four tomato varieties to salt stress. To see these differences, it was necessary to include in the study a variant treated with Atonik, but we were directly interested by the reaction of the tomato plants under the action of the Atonik biostimulant and the 120 Mm saline solution.

All the revisions to manuscript were marked up using the “Track Changes”.

At the editor's recommendation, we made the changes in the modified manuscript from the point of view of the English language.

The details about the application of the Atonik biostimulant were not included in the last version to the manuscript, I will do so with your permission.

Please address all correspondence concerning this manuscript to me at miha_bologa@yahoo.com.

Thank you for your consideration of this manuscript.

Sincerely,

Mihaela Covașă, PhD

Assistant Professor, Department of Plant Science

IULS Iasi

0749622234

Co-author

Alina Elena Marta, PhD

Lecturer, Department of Plant Science

IULS Iasi

alinamarta_fiziologie@yahoo.com

Cristina Slabu, PhD

Lecturer, Department of Plant Science

IULS Iasi

cristinaslabu@yahoo.com

Carmenica Doina Jităreanu

Professor, Department of Plant Science

IULS Iasi

doinaj@uaiasi.ro

Round 2

Reviewer 2 Report

Dear Sir/Madam

I appreciate your answer.

Regards

Author Response

January, 6, 2023

Dear, Reviewer

Thank you very much for the valuable suggestions you gave us to improve the quality of the article: Increasing the salt stress tolerance of some tomato cultivars under the influence of growth regulators “.

All the revisions to manuscript were marked up using the “Track Changes”.

Please address all correspondence concerning this manuscript to me at miha_bologa@yahoo.com.

Thank you for your consideration of this manuscript.

Sincerely,

Mihaela Covașă, PhD

Assistant Professor, Department of Plant Science

IULS Iasi

0749622234

Co-author

Alina Elena Marta, PhD

Lecturer, Department of Plant Science

IULS Iasi

alinamarta_fiziologie@yahoo.com

Cristina Slabu, PhD

Lecturer, Department of Plant Science

IULS Iasi

cristinaslabu@yahoo.com

Carmenica Doina Jităreanu

Professor, Department of Plant Science

IULS Iasi

doinaj@uaiasi.ro

Reviewer 3 Report

The authors answered all the questions raised. We reinforce that all these notes and clarifications indicated in the response letter must be inserted in the manuscript. This will reinforce the scientific robustness of the research carried out.

Author Response

January, 6, 2023

Dear, Reviewer

Thank you very much for the valuable suggestions you gave us to improve the quality of the article: Increasing the salt stress tolerance of some tomato cultivars under the influence of growth regulators “.

As you suggested, I have included in the article all the information presented in the previous letter.

All the revisions to manuscript were marked up using the “Track Changes”.

Please address all correspondence concerning this manuscript to me at miha_bologa@yahoo.com.

Thank you for your consideration of this manuscript.

Sincerely,

Mihaela Covașă, PhD

Assistant Professor, Department of Plant Science

IULS Iasi

0749622234

Co-author

Alina Elena Marta, PhD

Lecturer, Department of Plant Science

IULS Iasi

alinamarta_fiziologie@yahoo.com

Cristina Slabu, PhD

Lecturer, Department of Plant Science

IULS Iasi

cristinaslabu@yahoo.com

Carmenica Doina Jităreanu

Professor, Department of Plant Science

IULS Iasi

doinaj@uaiasi.ro